# Physics-Based Signal Analysis of Genome Sequences: An Overview of GenomeBits

**DOI:** 10.3390/microorganisms11112733

**Published:** 2023-11-09

**Authors:** Enrique Canessa

**Affiliations:** The Abdus Salam International Centre for Theoretical Physics (ICTP), 34151 Trieste, Italy; canessae@ictp.it

**Keywords:** genome sequence, SARS-CoV-2, comparative genomics variants, alternating series

## Abstract

A comprehensive overview of the recent physics-inspired genome analysis tool, GenomeBits, is presented. This is based on traditional signal processing methods such as discrete Fourier transform (DFT). GenomeBits can be used to extract underlying genomics features from the distribution of nucleotides, and can be further used to analyze the mutation patterns in viral genomes. Examples of the main GenomeBits findings outlining the intrinsic signal organization of genomics sequences for different SARS-CoV-2 variants along the pandemic years 2020–2022 and Monkeypox cases in 2021 are presented to show the usefulness of GenomeBits. GenomeBits results for DFT of SARS-CoV-2 genomes in different geographical regions are discussed, together with the GenomeBits analysis of complete genome sequences for the first coronavirus variants reported: Alpha, Beta, Gamma, Epsilon and Eta. Interesting features of the Delta and Omicron variants in the form of a unique ‘order–disorder’ transition are uncovered from these samples, as well as from their cumulative distribution function and scatter plots. This class of transitions might reveal the cumulative outcome of mutations on the spike protein. A salient feature of GenomeBits is the mapping of the nucleotide bases (A,T,C,G) into an alternating spin-like numerical sequence via a series having binary (0,1) indicators for each A,T,C,G. This leads to the derivation of a set of statistical distribution curves. Furthermore, the quantum-based extension of the GenomeBits model to an analogous probability measure is shown to identify properties of genome sequences as wavefunctions via a superposition of states. An association of the integral of the GenomeBits coding and a binding-like energy can, in principle, also be established. The relevance of these different results in bioinformatics is analyzed.

## 1. Introduction

Bioinformatics methods used to identify genome signatures usually associate some numerical encoding to strands of symbolic nucleotide letters, such as (A)denine, (C)ytosine, (G)uanine and (T)hymine (or (U)racil nucleobase in the nucleic acid RNA). This implies that nucleotides are assigned some fixed numerical values for the purpose of extracting information. As a consequence of this mapping, some features of the original sequence have been derived, which are preserved when the associated numerical representation is unique. In general, the properties of large datasets correlate a genome sequence when there is an absence of mapping function degeneracy.

For example, the well-known binary indicator of Voss [1] converts a DNA sequence to four one-to-one sequences of 0 and 1 to represent the characters A,T,C and G, correspondingly. Signal processing methods such as discrete Fourier transform (DFT) Power Spectrum and iterative graphical representations of DNA have been applied by assigning the numerical sequences (0,1) [2]. The goal is to reconstruct the DNA sequence at any point and to display genome signatures of the whole sequence starting from these partial indicators. This is relevant to compare genomics sequences when only parts of a genomics strand are available. These alignment-free comparison approaches have been extensively investigated in the last decade, and most recently for SARS-CoV-2 sequences [3].

Motivated by these simple numerical (0,1) representations of biological sequences, and in response to the urgent need of research on SARS-CoV-2 during the global pandemic years 2020–2022, we introduced a new numerical mapping algorithm, named GenomeBits, to unravel intrinsic signals from coronavirus (hCoV-19) sequences in FASTA format. In a series of papers written during the pandemic, interesting GenomeBits findings for different pathogen variants were reported in several references [4,5,6,7]. These investigations are now summarized in this overview with new examples to demonstrate the utility of this alternative bioinformatics method.

The significance of GenomeBits is to surpass some limitations of other statistical procedures, which seldom support a focused analysis at each single nucleotide level. Alternative frameworks for handling genomics information in the strands of DNA and RNA molecules from pattern matching consider symbolic codons (i.e., triplets of nucleotides) sometimes through manual examination. State-of-the-art methods for genomics analysis to identify variations and characteristic profiles still need algorithms to compare complete genome sequences quickly, accurately and efficiently [8]. In this light, GenomeBits might become another feather in bioinformatics toolkits for genome characterization. The GenomeBits approach has, in practice, a twofold advantage: first, it allows us to explore possible short-range trends and local features of complex adenovirus sequences. The discovery of genomic signals at “small” scales— for instance, of 30,000 base position (bp)—is potentially significant to search relevant genomic signals and detect general trends and basic properties. The second salient feature of the GenomeBits approach is that it provides insights for each nucleotide bases A,C,G and T separately, adds capabilities for fast sequence taxonomy and easy data visualization. Our hope is to encourage researchers to develop and apply physics-inspired analysis tools, like GenomeBits, based on traditional signal processing methods to extract underlying genomics features from the distribution of nucleotides and to compare mutation patterns in viral genomes.

The article is structured as follows. Section 2 describes in great detail the GenomeBits method. We discussed how the nucleotide bases A,T,C and G are mapped as a spin-like numerical sequence via a finite alternating sum series having a distribution of (0,1) indicators for each A,T,C and G. This expansion of the genome sequences from 1D to 4D leads to the evaluation of computationally inexpensive statistical distribution curves. Section 3 displays the results of using the Fourier transform of SARS-CoV-2 genomes in different geographical regions. These include GenomeBits analysis for complete genome sequences for the first coronavirus variants reported: Alpha, Beta, Gamma, Epsilon and Eta. Interesting features of the other coronavirus variants Delta and Omicron in the form of a unique ‘order–disorder’ transition is uncovered from these samples, as well as their cumulative distribution function (CDF) and scatter plots. Section 4 introduces the quantum-inspired GenomeBits model extension. Such an analogous probability measure allows us to identify emergent properties of genome sequences in the form of wavefunctions via a superposition of states. The final section (Section 5) concludes this review with a possible association of the integral of the GenomeBits coding and a binding-like energy, and proposes some future directions for research.

## 2. GenomeBits Mapping

The GenomeBits mapping to retrieve special patterns of complete genome sequences assigns alternating sum series having multiple (0,1) values for the nucleotide variables α=A,C,T,G of genome sequences [4]. The mapping of letter sequences into binary values is not new, as already mentioned [1]. The novelty in the numerical encoding of GenomeBits lies in the use of positive and negative signs ${\left(-1\right)}^{k-1}$ in the *k*-base sums. The analysis of genomics sequencing throughout finite alternating sums allows us to extract distinctive features at each nucleotide base. From a statistics point of view, these alternating series correspond to a time series of discrete values.

The arithmetic progression in question carries, in fact, alternating terms, and a finite positive moment of independently distributed variables $X_{k,\alpha }$, as follows:(1)$$E_{\alpha ,N}\left(X\right)=\sum_{k=1}^{N}{\left(-1\right)}^{k-1}X_{\alpha ,k}\;.$$

The individual terms $X_{k}$ are associated to 0 or 1 values according to their base position along the genome sequences of length *N*. They are assumed to satisfy the relation:(2)$$X_{\alpha ,k=N}=\left|E_{\alpha ,N}\left(X\right)-E_{\alpha ,N-1}\left(X\right)\right|\;.$$

In this GenomeBits mapping, the binary indicator for the sequences assumes alternating positive and negative signs. The (±) signs are assigned sequentially, starting with +1 at k=1 as in the example in Table 1. If a term $X_{\alpha ,k}$ is positive or zero (1 or 0) at a given nucleotide bp *k*, then the next $X_{\alpha ,k+1}$ term can be negative or zero again (−1 or 0). This selection is inspired by the discrete physics Ising spin model, in which variables can be in one or more states with spin up or spin down in a lattice (representing magnetic dipole moments of atomic spins). It can also be interpreted in terms of a balanced ternary logic, which uses the three digits −1, 0 and +1. Balanced ternary has many applications, including the unit of quantum information realized by a three-level quantum system (qutrit or quantum trit), that may be in a superposition of mutually orthogonal quantum states—just as the qubit.

A GenomeBits mapping example of the hCoV-19 genome fragment GTATACTGCTGC (having 12 nucleotides) converted to the alternating (0,1) array via Equation (Equation 1) is as follows.

On the other hand, a quantum-based extension of our GenomeBits approach to retrieve properties of genome sequences converted to 0 and 1 outcomes can also be derived from the above class of series [7]. In this way, GenomeBits can become a measurement theory and not a model of quantum physical processes at the atomic level. This “analogous” extension to (real and imaginary parts of longitudinal) wavefunctions represents a mathematical description of an isolated, “analogous” quantum system where the wavefunction vs. the nucleotide bases present features of sound waves. This approach leads to reveal novel measures of the genome evolution at nucleotide levels during genome mutations over *N* intervals.

Let us consider Equation (Equation 1) as the resulting wave generated by a certain superposition of multiple discrete wavefunctions, i.e., $\phi \left(X_{\alpha ,N}\right)⇌\psi_{n}\left(X_{\alpha ,N}\right)$ in a medium. In polar form, we then extend the GenomeBits approach to write
(3)$$\begin{array}{ccc} \psi_{n}\left(X_{\alpha ,k}\right) & \equiv & A\;{\left(-1\right)}^{k-1}\left|\Phi \left(X_{\alpha ,k}\right)-\Phi \left(X_{\alpha ,k-1}\right)\right|\;\exp\left\{\frac{n\pi i}{\lambda_{{}_{N}}}\Phi \left(X_{\alpha ,k}\right)\right\}\;,\\ & = & A\;{\left(-1\right)}^{k-1}X_{\alpha ,k}\;\exp\left\{\frac{n\pi i}{\lambda_{{}_{N}}}E_{\alpha ,k}\left(X\right)\right\}\;, \end{array}$$
where *A* is a real constant and n=1,2,…. These wavefunctions are complex functions in general and the displacement of the wave is a function of the *k*-bp. The normalization condition via the complex conjugate $\sum_{k=1}^{N}{\left|\psi_{n}\left(X_{\alpha ,k}\right)\right|}^{2}=\sum_{k=1}^{N}\psi_{n}\left(X_{\alpha ,k}\right)\psi_{n}^{*}\left(X_{\alpha ,k}\right)=1$ leads to
(4)$$A^{2}\sum_{k=1}^{N}{\left|{\left(-1\right)}^{k-1}X_{\alpha ,k}\right|}^{2}=A^{2}\sum_{k=1}^{N}X_{\alpha ,k}^{2}=1\;.$$

Since $X_{\alpha ,k}$ takes (0,1) values only, the amplitude then satisfies
(5)$$A=\frac{1}{\pm \sqrt{N_{{}_{+1}}}}\;,$$
where we denote by $N_{{}_{+1}}$ the total number of 1’s found in a complete $N=N_{{}_{0}}+N_{{}_{+1}}$ sequence within each species α. The maximum real value for the alternating sum of binary sequences of Equation (Equation 1) can be obtained by the choice:(6)$$\lambda_{{}_{N}}\equiv \sum_{j=1}^{N}\phi \left(X_{\alpha ,j}\right)=\sum_{k=1}^{N}{\left(-1\right)}^{k-1}X_{\alpha ,k}\equiv E_{\alpha ,N}\left(X\right)=\Phi \left(X_{\alpha ,N}\right)\;.$$

From Equation (Equation 3) and Euler’s identity, we note that for k→N and ∀n≥1, the Cartesian form of ψ oscillates and decreases for large numbers of 1’s, namely
(7)$$\psi_{n}\left(X_{\alpha ,N}\right)=\left(\frac{1}{\pm \sqrt{N_{{}_{+1}}}}\right){\left(-1\right)}^{N-1}X_{\alpha ,N}\;\left[\cos\left(n\pi \right)+i\;\sin\left(n\pi \right)\right]=\frac{{\left(-1\right)}^{n}}{\pm \sqrt{N_{{}_{+1}}}}\;\phi \left(X_{\alpha ,N}\right)\;.$$

Hence, the GenomeBits Equation (Equation 1) for complete genomics strands can be seen as a steady state wave created by some non-zero complex wavefunctions ψ. The wavefunction changes with k≠0 having maximum density probability $1/N_{{}_{+1}}$ at each *n*-value.

## 3. Discussion

The complete genome sequences of the contagious SARS-CoV-2 coronavirus in humans were first reported by Chinese scientists in early 2020 [4]. To this date, there are thousands of sequence sources openly available through the well-known GISAID Initiative (www.gisaid.org, accessed on 11 November 2023) that allowed for comparisons and classifications of the species of different emerging coronavirus variants and genome mutations from around the globe during all of the pandemic years 2020–2022. These data were extremely useful for genomics surveillance and, today, they form a rich source of information for the development of new theories that may predict and catalog this class of genomics strands. For almost half of its genome, the coronavirus presented a unique lineage with only a few relationships to other known viruses, in particular, in the spike region encoding the S-protein.

### 3.1. DFT Power Spectrum

Using GISAID data, we presented in [4] the GenomeBits signal analysis of Equation (Equation 1) for the complete genome sequences for coronavirus variants B.1.1.7 (Alpha), B.1.135 (Beta), P1 (Gamma), B.1.429–B.1.427 (Epsilon) and B.1.525 (Eta), with *N* nucleotides on the order of 30k bp in length. These samples were taken from broad geographical regions and were collected during different periods of the pandemic starting from November 2020 up to March 2021. In [4], we discussed how the GenomeBits method provides additional information to conventional similarity comparisons via alignment methods and DFT Power Spectrum approaches. To obtain these results, a simple graphics user interface (GUI) was implemented, which can be downloaded from Github [9]. The Linux-based GUI runs under Ubuntu O.S. and requires little processing time for the analysis of complete genomics data. The GenomeBits GUI considers samples with A,C,T,G sequences corresponding to genomics sequence data from given variants and different countries. Sequences uncompleted with codification errors (for example, ‘NNNN’ letters) are not considered.

As mentioned, GenomeBits can be used to investigate the conventional DFT Power Spectrum to identify the base periodicity properties of genome sequences and classification variants during the pandemic. In general, the Power Spectrum of the microorganism sequence is usually considered as the sum of the partial spectra: $\sum_{\alpha }|S_{\alpha }{\left(f\right)|}^{2}=\left(1/N^{2}\right)\sum_{\alpha }^{N}{\left|X_{\alpha ,k}\;exp\left(2\pi ifk\right)\right|}^{2}$, with discrete frequencies f=1/N,2/N,…. The DFT maps genome sequences into four (0–1) indicators and evaluates them in the frequency domain. In our method, DFT can provide some insights for single-nucleotide bases A,C,G and T to characterize virus variants, as illustrated in Figure 1, for nucleotide bases from different variants and countries (the same as in [4]).

As can be seen in the plots, the DFT of the alternating sum series of Equation (Equation 1) reveals partial peaked structures appearing at a ‘frequency’ equal to 16.66, implying a 50/3 characteristic periodicity, each above noise level, for the alternating sums. We note that larger peaks appear in the nucleotides of the complementary strands T and G only. This interesting result can be detected only in the frequency domain. In [4], it was shown that their total sum $|S_{A}{|}^{2}+|S_{T}{|}^{2}+|S_{C}{|}^{2}+{\left|S_{G}\right|}^{2}$ has a peak at around the frequency 33.33 for all variants considered. This is a unique, distinctive pattern retrieved from the intrinsic data organization of genome sequences according to their progression along the nucleotide bp.

### 3.2. ‘Order–Disorder’ Transition

The GenomeBits Equation (Equation 1) can also reveal another interesting underlying genomics feature of coronavirus variants, namely an ‘order–disorder’ transition [5]. In Figure 2, we show results for the sequences of the coronavirus variants AY.4.2 Delta and B.1.1529 Omicron samples from Spain, together with their average values, as indicated. The Delta variant was first detected in India at the end of 2020 and then in South Africa in late 2021 to then become dominant world-wide. Delta and Omicron variants had common mutations in the building blocks that conform the spike protein (responsible for the pathogen penetration in a human cell). Omicron caused a less severe COVID-19 infection than the Delta variant during the pandemic waves. Nature selected Omega mutations, which seemed to replicate more efficiently.

On the left of Figure 2, the results for the pair nucleotides A,C are shown, and on the right are the complementary nucleotides T,G. In the figure, we found regions where the alternating sums from the Delta data (in orange) mirror those of the Omicron (in gray). This behavior is highlighted by the green values after averaging both curves. The regions having low data noise (or rather constant averaged values) indicate a coding correspondence between variants, especially around the coronavirus S-spike region. These results reveal a sort of ‘disordered’ (or peaked)-to-‘ordered’ (or constant) transition, indicated by the red arrows for each single nucleotide level. This may reflect the polymorphism in the genomics variations by only assigning alternating (0,1) values to symbolic nucleotide characters.

It is important to observe the distinctive Guanine patterns for Delta and Omicron variants of coronavirus samples from Spain, as shown on the top right of Figure 2b. The Guanine ‘ordered-to-disordered’ GenomeBits transition appears to be different from the other three nucleic acid bases A,C and T. At the arrow-pointed region, the Guanine transition appears to have been inverted. Larger (‘disordered’) peaked structures are found inside the S-spike region in contrast with those for A,C and T, which appear outside the S-spike region, displaying a more ‘ordered’ (or constant) transition (i.e., Figure 2a,c,d). This class of profile might reveal the cumulative outcome of mutations on spike protein from Delta to Omicron coronavirus variants by some A,C or T > G substitutions or vice versa. Significant differences in the spike region can impart these unique properties found for Guanine. The E484 SARS-CoV-2 spike protein mutation corresponds to a nucleotide substitution from Guanine (as in the original Wuhan-Hu-1 isolate) to Adenine (Beta and Gamma variants) or Cytosine (Kappa variants) (see [10]). Alongside this, the observed ‘disordered’ transitions in the non-spike regions for A,C,T curves (displayed around 10,000–20,000 base positions) could also be responsible elements behind single-nucleotide mutations, which might lead to properties, including growth advantages, transmission rate and immune evasion, among others.

### 3.3. Statistical Imprints

The evolution of other infectious disease related to Monkeypox virus (MPXV) also alerted humanity in 2022. Soon after, the genome sequences for MPVX were analyzed using the GenomeBits method [6]. In that study, histograms, empirical and theoretical cumulative distribution curves and the resulting scatter plots for the base nucleotides A−C versus their complementary base nucleotides T−G for MPVX were reported. In the following, we extend these statistical investigations and apply them anew to the signal analysis of typical hCoV-19 genome sequences of lineage B.1.1.529 Omicron from the USA (ID EPI_ISL_7887528 and EPI_ISL_7887531).

In Figure 3, we show the histogram curve (i.e., number of data points in a given bin), a standard CDF and the scatter plot obtained from two different histogram datasets of equally sized bins. In the figure, the blue and green bar graphs are the histograms and the red full line and red dotted lines are empirical and theoretical CDF curves, respectively. In the computations, 50 bins are used, and the blue line fits have been obtained via a Gaussian distribution.

The resulting scatter diagrams for A−C versus T−G base nucleotides allow us to highlight correlations and anomalies at certain colored contact points between GenomeBits sequences of complementary nucleotide bases. The clearer colors of the hexagon markers add a new dimension to the study of patoghens. In fact, the plotted patterns allow for visual comparisons. Markers show how the mapped binary data are stratified along the sequences and how correlated points fall along distinct shapes. The patterns found are unequally distributed in the scatter plot and these do not occur randomly.

In this overview, we considered the distinctive patterns of the complete full letter sequence of all the genes of the coronavirus genome for single strands, as available from, and reported within, the GISAID Initiative for the building blocks of nucleic acid coding sequence Adenine, Cytosine, Guanine and Thymine. In RNA, Thymine is replaced by Uracil because of their six-member single-ring structural similarity referred to as pyrimidines. The chemical structures of Uracil and Thymine, composed of carbon and nitrogen atoms, are very similar. The presence of Thymine in a DNA strand in actively dividing cells is more stable thermodinamically and improves the efficiency of DNA replication when compared to Uracil in RNA. The complementary base of both Uracil and Thymine is Adenine. During the pandemic, it was observed that SARS-CoV-2 genomes contain more Uracil than any other nucleotide. The substitution of nucleotides to Uracil was the highest among the non-synonymous mutations observed.

From these simple statistical imprints, we believe that the GenomeBits method can help to shed light behind the behavior of infectious diseases by focusing on single-nucleotide structures—structures which are different on each mutation of a virus.

## 4. Wave-like Features

Acoustic-like sounds within a biological context can help to identify trends in genome sequences and characterize new properties. For example, sonification algorithms based on biological rules for DNA sequences have used codons to generate audio strings representative of synthesized RNA during transcription [11]. Another possibility to identify emergent properties of genome sequences in the form of complex wavefunctions can be obtained using Equation (Equation 3). This wavefunction becomes a mathematical description of an analogous quantum system.

In this context, it is worthy to note that at the initial base position k=1, the following relations follow from our previous equations
(8)$$\begin{array}{ccc} |\psi_{n}\left(X_{\alpha ,1}\right){|}^{2} & = & \psi_{n}\left(X_{\alpha ,1}\right)\;\psi_{n}^{*}\left(X_{\alpha ,1}\right)\;,\\ & = & a_{1}^{2}{\left(\Psi_{n}^{\left(a\right)}\left(X_{\alpha ,1}\right)\right)}^{2}+b_{1}^{2}{\left(\Psi_{n}^{\left(b\right)}\left(X_{\alpha ,1}\right)\right)}^{2}\;,\\ & = & \frac{1}{N_{{}_{+1}}}\;X_{\alpha ,1}^{2}\;, \end{array}$$
and also
(9)$$\begin{array}{ccc} b_{1}\Psi_{n}^{\left(b\right)}\left(X_{\alpha ,1}\right) & = & a_{1}\Psi_{n}^{\left(a\right)}\left(X_{\alpha ,1}\right)\;tan\left(\frac{n\pi }{\lambda_{{}_{N}}}X_{\alpha ,1}\right)=\frac{1}{\pm \sqrt{N_{{}_{+1}}}}\;X_{\alpha ,1}\;sin\left(\frac{n\pi }{\lambda_{{}_{N}}}X_{\alpha ,1}\right)\;,\\ a_{1}\Psi_{n}^{\left(a\right)}\left(X_{\alpha ,1}\right) & = & \frac{1}{\pm \sqrt{N_{{}_{+1}}}}\;X_{\alpha ,1}\;cos\left(\frac{n\pi }{\lambda_{{}_{N}}}X_{\alpha ,1}\right)\;. \end{array}$$

This means that the wavefunction $\psi_{n}$ can be, in principle, further divided into two wavefunctions $\Psi_{n}^{\left(a\right)}$ and $\Psi_{n}^{\left(b\right)}$. This may lead to an analysis of spatial stereo sonification from genome sequences.

The applicability of this approach to sound waves from complete genome sequences of coronavirus pathogens was discussed in [7]. At the level of nucleotide ordering for different genome mutations, this description of the genome dynamics revealed interesting results. As shown next, the real (and also imaginary) parts of the longitudinal wavefunction vs. the nucleotide bases display characteristic features of sound waves starting with inputs of ones and zeros only.

As shown by the illustrative results in Figure 4, the real spectrum of $\psi_{n=1}$ from Equation (Equation 7) vs. the nucleotide base positions of the 2020 coronavirus Wuhan-Hu-1, China (ID MN908947), presents typical features of sound waves. At any point different from zero, binary projections at the level of nucleotides display oscillatory patterns in correspondence with the intrinsic gene organization. This behavior is a consequence of factorizing the discrete wavefunction as the product of a complex exponential function and a linear function proportional to the (0,1) sequences. These wavefunctions can be seen as standing waves with zero nodes. They are an analogous mathematical construction based on physics quantum states with some total physics energy. Such analogous wavefuntion calculations from genome sequences can be applied to sonification as in [7] and illustrated next.

The analogous time–frequency spectrogram of the nucleotide base of Adenine, showing large peak signals over time, is shown in Figure 5—generated via separated GenomeBits wavefunction projections of nucleotide bases A,C,G and T. As in [7], data points in the audio curves are fitted to a Gaussian in order to obtain a more continuous audio spectrum in the waveform reconstruction of the Wuhan-Hu-1 coronavirus sequence (ID MN908947.3). In these calculations [7], we use a similar sample rate of 4096, with precision: 16-bit; duration: 2:46.28 min for 681,097 sampling; file size: 1.36 M; bit rate: 65.5 k and sample encoding in one channel: 16-bit.

A visual inspection of the spectrograms obtained from an audio file could help to identify significant virus mutations. Examples of the wav audio files generated via the present extended GenomeBits wavefunctions, by transforming the occurrence of nucleotides of the same class along the genome sequences, can be downloaded from GitHub [12]. We have artificially shifted these curves in frequency by an offset in Hz to obtain a clearer picture when assessing significant variations through, e.g., the different densities of colored straight lines. This frequency shift of the analogous audio data simply allows us to hear the chirp signal better.

## 5. Final Remarks and Future Directions

This overview summarizes the GenomeBits findings in [4,5,6,7], outlining the intrinsic signal organization of genomics sequences for different coronavirus variants during the pandemic years 2020–2022. All results discussed are representative and the plots shown are calculated anew. The DFT Power Spectrum data were calculated for Delta and Omicron variants for coronavirus samples from Spain. A difference with respect to Ref. [4] is that these curves represent results obtained for single A,C,G and T nucleotides and not their total sum as previously published. The results obtained for a ’order–disorder’ transition are for each sequence of A,C,G and T nucleotide of the coronavirus Delta and Omicron variants from Spain (a different sequence was used in Ref. [5]). In Ref. [6], the statistical imprint calculations showed a relation with the MPXV disease. In this review, completely new results are reported for histograms, empirical and theoretical cumulative distribution curves and the resulting scatter plots for the base nucleotides A–C versus their complementary base nucleotides T–G for hCoV-19 genome sequences of Omicron from the USA. The real spectrum of the analogous wavefunction vs. the nucleotide base positions, corresponding to the 2020 coronavirus Wuhan-Hu-1, China, behave like sound waves (shown in Figure 4 and Figure 5). The previous Ref. [7] reports results for representative nucleotide bases from the genome sequence of the coronavirus Omicron variant. Therein, such curves were artificially shifted in frequency by an offset of 400 Hz.

In this work, we have considered distinctive patterns of a complete coronavirus genome code for a single strand folded onto itself, as available from the GISAID database, for the nucleotide bases Adenine, Cytosine, Guanine and Thymine. As we discussed in [4], the coronavirus genome is RNA, not DNA. It is possible to correlate symbols used for proteins (polymers of amino acids) to that of nucleic acids (polymers of nucleotides). Genomebits’ numerical results may be relevant to assist in designs of the new generation of synthetic messenger ribonucleic acid (mRNA)-based vaccines through a continuous surveillance of the evolution of sequence mutations and their capability to replicate. Indirectly, such studies could relate to the “central dogma” of modern molecular biology via characterizing the processes involved in transferring genetic code from DNA into proteins through mRNA. This is so because inter-gene parts of a sequenced genome (revealed potentially by GenomeBits data mine) play most likely important roles in the transcription and translation of protein synthesis.

To conclude this comprehensive bioinformatics overview of the underlying genomics features at the nucleotide level from the viewpoint of the physics-based GenomeBits approach, let us mention potential directions to explore in order to enrich the description of data sequencing. In theory, one can consider the GenomeBits progression $E_{\alpha ,N}\left(X\right)$, carrying alternating terms of the independently distributed (0,1) variables $X_{\alpha ,k}$, as the function that contains (most of) the hidden information (or the “secrets”) of the genome (en)coding. Processes of binding should be energetically optimized and force driven. If one were to accept this interpretation, then by some simple geometrical and physics reasoning, it can be argued that along a 1D continuous positive string, the area under a given coding curve f(x) satisfies
(10)$$\int f\left(x\right)dx=\zeta \;\frac{\Delta x}{F}\;\;\;,$$
where ζ is energy, *F* is applied force and Δx is the displacement along our GenomeBit system. By considering a spring-like force $F=-K_{{}_{c}}\cdot \Delta x$, we then approximate for simplicity:(11)$$\zeta_{{}_{b}}\approx -K_{{}_{c}}\int f\left(x\right)dx\;\;\;.$$

This relation may correlate a negative binding energy $\zeta_{{}_{b}}$ with the properties contained in the “information” coding experienced through a (discrete “force” series) function $f\left(x\right)\Leftrightarrow E_{\alpha ,k}$ of Equation (Equation 1) of a supporting genetic sequence (the “system” following the information hypothesis of the theoretical framework in Ref. [13]). Instructions to assemble all living organisms must be implicitly included in the GenomeBits series and the area under these GenomeBits curves can relate to an energy. In other words, the energy–information correlation of Equation (Equation 11) may provide some hints to describe the evolution of nucleotides located in the cell nucleus—the fundamental unit responsible for life. The genome in a cell carries fundamental information to create a living organism, and part of this embedded information is established through meaning and symbols.

All associations discussed in this review are likely to be relevant for the study of genomics encoding of new sequences in microorganisms. Consequently, the GenomeBits representation may be useful to target the evolution of sequences due to natural mutation. One advantage for this approach is that sequence data for single A,T,C and G can be handled statistically to find their diverse characterizations and determine if the altered coding regions (genes) may share similar behaviors to those archived and classified during the evolution of a pandemic. GenomeBits may be useful for the bioinformatics surveillance behind future infectious diseases by means of its simple letter sequence-to-alternating numerical mapping.

## Figures and Tables

**Figure 1 microorganisms-11-02733-f001:**
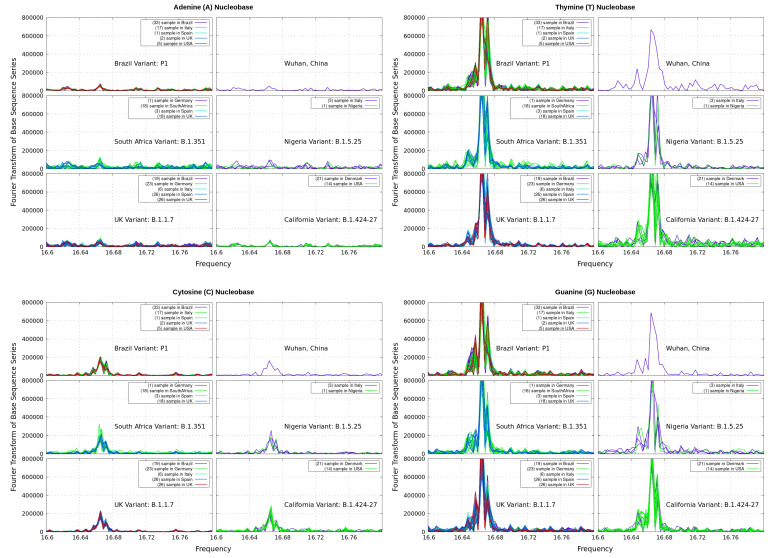
Discrete Fourier transform of the alternating sum series of Equation (Equation 1) for nucleotide bases according to its progression *N* along different samples of coronavirus genome available from different countries.

**Figure 2 microorganisms-11-02733-f002:**
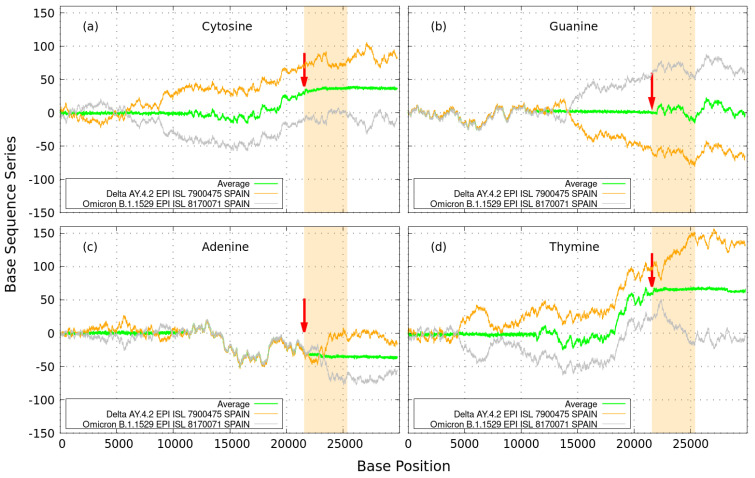
GenomeBits sum series: Delta (in orange) and Omicron (in gray) variant imprints for coronavirus samples from Spain. Arrows show a ‘disordered’ (peaked)-to-‘ordered’ (constant) transition around the S-spike region of the SARS-CoV-2 Wuhan-Hu-1 sequence, shown in clear red. At the arrow-pointed region of Guanine in subfigure (**b**), larger ‘disordered’ structures are found inside the S-spike region in contrast with those for A,C and T (i.e., subfigures (**a**,**c**,**d**)), which appear outside the S-spike region, displaying a more ‘ordered’ transition.

**Figure 3 microorganisms-11-02733-f003:**
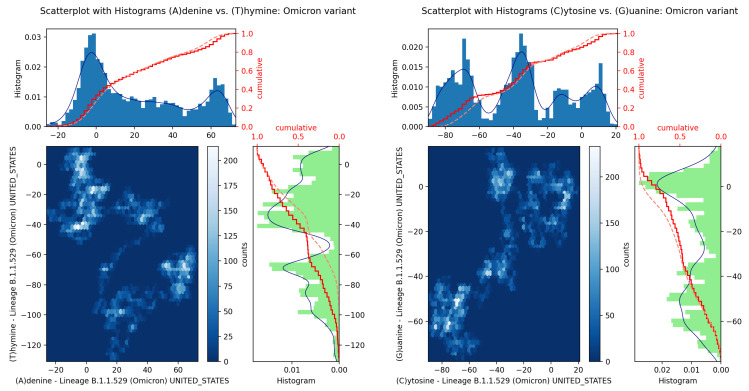
Scatter plot of GenomeBits Adenine and Thymine curves with histograms from the hCoV-19 genome sequences of lineage B.1.1.529 Omicron from the USA (ID EPI_ISL_7887528 and EPI_ISL_7887531). Red full line and red dotted lines are empirical and theoretical CDF curves, respectively. Blue lines fit Gaussian distributions.

**Figure 4 microorganisms-11-02733-f004:**
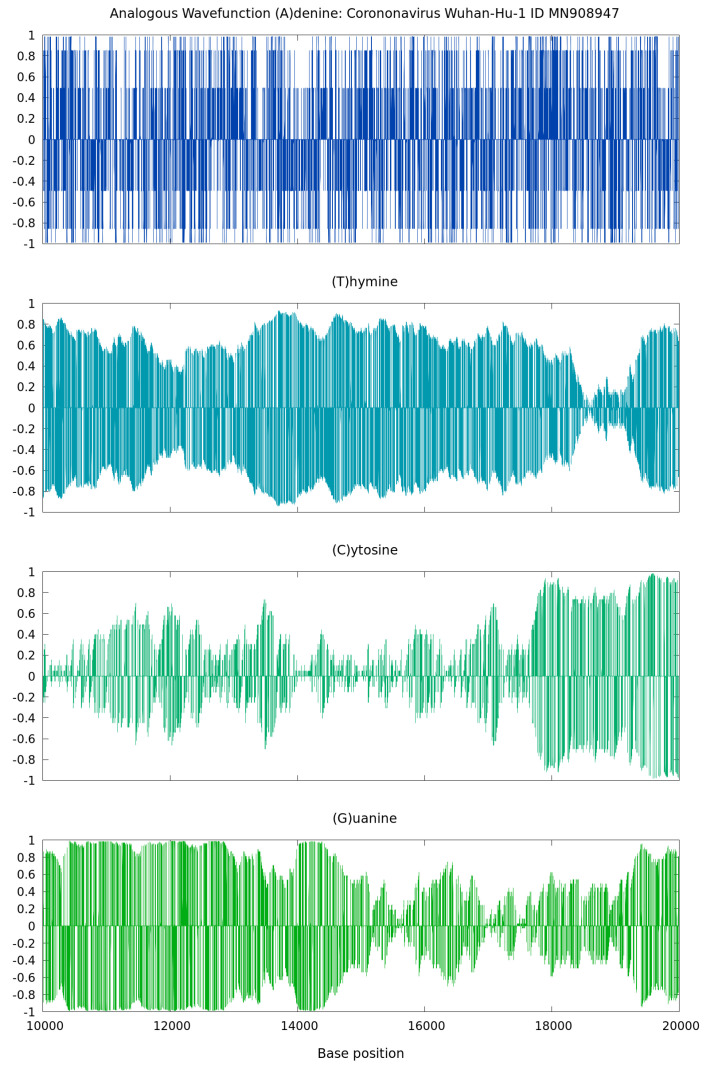
Real part of the wavefunction in $\psi_{n}$ for n=1 derived using the GenomeBits method.

**Figure 5 microorganisms-11-02733-f005:**
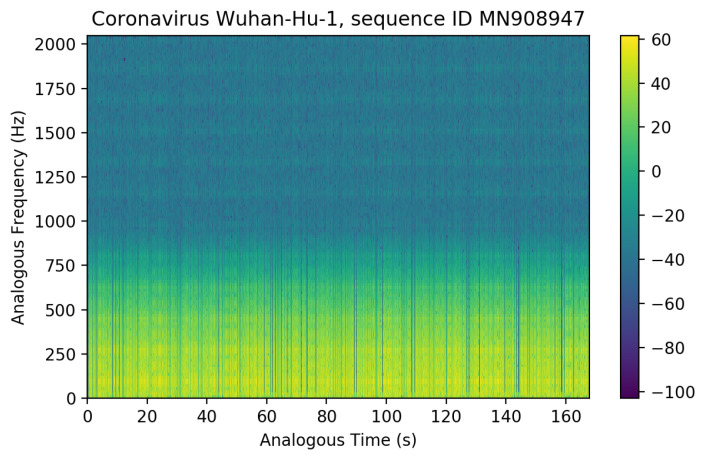
Analogous time–frequency spectrogram of nucleotide base Adenine produced from a wav audio file generated via the GenomeBits wavefunction from the first Wuhan-Hu-1 coronavirus sequence.

**Table 1 microorganisms-11-02733-t001:** GenomeBits mapping via Equation (Equation 1) for N=12.

Base Position *k*	1	2	3	4	5	6	7	8	9	10	11	12	GenomeBits Sums
Sequence (strand)	**G**	**T**	**A**	**T**	**A**	**C**	**T**	**G**	**C**	**T**	**G**	**C**	$\sum_{k=1}^{12}{\left(-1\right)}^{k-1}X_{\alpha ,k}$
${\left(-1\right)}^{k-1}X_{\alpha =A,k}$	0	0	+1	0	+1	0	0	0	0	0	0	0	+2
${\left(-1\right)}^{k-1}X_{\alpha =C,k}$	0	0	0	0	0	−1	0	0	+1	0	0	−1	−1
${\left(-1\right)}^{k-1}X_{\alpha =G,k}$	+1	0	0	0	0	0	0	−1	0	0	+1	0	+1
${\left(-1\right)}^{k-1}X_{\alpha =T,k}$	0	−1	0	−1	0	0	+1	0	0	−1	0	0	−2

## Data Availability

Binaries for “GenomeBits: A tool for the signal analysis of complete genome sequences”, https://github.com/canessae/GenomeBits/ (Last visited 28 October 2022).

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
