# Peer review of "Physics-Based Signal Analysis of Genome Sequences: An Overview of GenomeBits"

_microorganisms, 2023, doi:10.3390/microorganisms11112733_

Round 1
Reviewer 1 Report
Comments and Suggestions for Authors
The present manuscript reports an interesting study - Physics based signal analysis of genome sequences.
In my opinion, the abstract should provide more information about the contents of the manuscript. The Introduction can also be improved, including a statement on how GenomeBits can be another feather in Bioinformatics toolkit for genome characterization. Some parts from "Final Remarks and future directions" such as the equations etc can be moved to the Discussion section. The other sections are written clearly. I would suggest the author to read the manuscript for grammatical corrections.
Comments on the Quality of English Language
Minor editing of English language is required. Grammar and words can be improved at multiple places.
Reviewer 2 Report
Comments and Suggestions for Authors
This manuscript overviewed a genome analysis tool, GenomeBits, which is inspired by traditional signal processing methods such as DFT. GenomeBits can be used to extract genomics features from the distribution of nucleotides, and can be further used to analyze the mutation patterns in viral genomes. Examples of SARS-CoV-2 cases were presented to show the usefulness of GenomeBits. The manuscript is well-organized and interesting to readers. Some issues need to be fixed, though.
1. In Fig. 2, Guanine’s pattern (top right subfigure) is different from the other three: from ordered to disordered at the arrow-pointed region. This should be mentioned and explained.
2. In Fig. 3, only A-T and C-G contacts are shown. But for RNAs, G-U base pairs are very common, which should be also included.
Reviewer 3 Report
Comments and Suggestions for Authors
The article started as a review of Genomebits but the content showed more towards the summary of what was done using genomebits to conduct a comparison between different variants of SARS-Cov-2 as well as Monkeypox. The article is relatively confusing with regards to the aim of the article and thus the authors should improve on guiding the reader towards what they want to achieve with this article.
Here are also some other comments aside from the general one above:
1. "Data Availability Statement: Binaries for "GenomeBits: A tool for the signal analysis of complete 291 genome sequences", https://github.com/canessae/GenomeBits/ (Last visited 20/08/202)." the date has an error
2. The "soundwave" graph from the article is novel but would a shift in base pair cause a different "soundwave" graph making it difficult to compare?
3. Why is it that in Figure 2 (please label the graph with A, B, C, etc...), 3 graphs showed a different position of change (labeled with red arrow) but for bottom left graph it showed the position where the graph disperse (position of change)
Thank you
Round 2
Reviewer 3 Report
Comments and Suggestions for Authors
Thank you kindly explaining and correcting the article accordingly